# The Effects of a Physical Activity Online Intervention Program on Resilience, Perceived Social Support, Psychological Distress and Concerns among At-Risk Youth during the COVID-19 Pandemic

**DOI:** 10.3390/children9111704

**Published:** 2022-11-06

**Authors:** Michal Glaser, Gizell Green, Avi Zigdon, Sharon Barak, Gili Joseph, Adilson Marques, Kwok Ng, Itamar Erez-Shidlov, Lani Ofri, Riki Tesler

**Affiliations:** 1Department of Health Systems Management, School of Health Sciences, Ariel University, Ariel 4077625, Israel; 2Health Promotion & Wellbeing Research Center, Ariel University, Ariel 4077625, Israel; 3Department of Nursing, School of Health Sciences, Ariel University, Ariel 4077625, Israel; 4Disaster Research Center, Ariel University, Ariel 4077625, Israel; 5Department of Pediatric Rehabilitation, The Chaim Sheba Medical Center at Tel Hashomer, Ramat-Gan 5262000, Israel; 6Department of Physical Education, Faculty of Science, Seminar Kibbutzim College, Tel Aviv 6250769, Israel; 7Faculdade de Motricidade Humana, Centro Interdisciplinar para o Estudo da Performance Humana, Universidade de Lisboa, 1649-004 Lisboa, Portugal; 8School of Educational Sciences and Psychology, University of Eastern Finland, 70211 Kuopio, Finland; 9Physical Activity for Health Research Cluster, Department of Physical Education and Sport Sciences, University of Limerick, V94 T9PX Limerick, Ireland; 10Department of Physical Education, Wingate College, Netanya 4290200, Israel

**Keywords:** health promotion, online intervention program, resilience, social support, adolescents health, COVID-19 pandemic

## Abstract

Background: COVID-19 restrictions have led to social isolation affecting youth’s health, particularly at-risk youth. Objectives: We examined whether an online mentoring health intervention (OMHI) would strengthen characteristics that can prevent risky behaviors: resilience, perceived social support, psychological distress, and crisis concerns. Methods: Fifty-six secondary-school students participated, 27 in the intervention group and 29 in the control group (mean age 16.18, SD 0.83 vs. 16.62, SD 0.82, respectively). The study took place between March and August 2020. Results: The intervention group was less resilient pre-test, with similar resilience levels as the control group post-test. Intervention group participants presented a significantly higher crisis level pre- and post-test than the control group, as well as an increase in resilience (effect size = 1.88) and social support (effect size = 1.22), while psychological distress significantly decreased (effect size = −1.03). Both groups (intervention vs. control) predicted changes from pre-to-post test for resilience and crisis (adjusted R^2^ = 0.33, *p* = 0.001 and R^2^ = 0.49, *p* = 0.0001 respectively). Conclusions: OMHI participation was associated with improved resilience and social support, and decreased psychological distress, making it an effective strategy in health promotion for at-risk youth. An online intervention program combining mentoring in physical activity and interpersonal connections may constitute an effective health promotion strategy for at-risk youth, especially in times of crisis.

## 1. Introduction

Physical activity (PA) in young people has numerous benefits on physical health, mental health, and educational achievement [1]. Physical inactivity and sedentary behaviour (SB) are associated with the risks of overweight, obesity, and cardiometabolic complications [2]. Conversely, regular PA reduces such risks and is associated with preventing the onset of many chronic diseases as well as premature mortality [3].

At-risk youth are individuals who have been exposed to risk factors in childhood that correlate with negative physical, mental, and/or emotional outcomes in later life [4]. This population may have been exposed to negative social interactions or delinquency that led to risky behaviours such as cigarette smoking, drug abuse, alcohol consumption, and dropping out of the traditional school system [4,5]. At-risk youth are at even higher risk during crises and traumatic events, due to a complex combination of potentially negative family interactions, economic uncertainty, stress and anxiety, and with limited access to resources and support [6,7].

People differ widely in how they respond to challenges and difficulties [8]. One’s ability to withstand setbacks, adapt positively to change, and bounce back from adversity and stress is described as “resilience” [9], a complex, dynamic, and individualized process that involves internal strength and external resources [10]. Resilience is a crucial skill for at-risk youth in terms of coping with crises, traumatic events, and psychological distress [11,12]. Individual resilience and well-being have been found to be primary predictors of COVID-19-related anxiety [13], as well as a protective factor for youth during the COVID-19 crisis [12].

Risk factors exist at the family level (e.g., discord, domestic violence, parental physical abuse, mental illness, economic deprivation) [14], community level (e.g., poverty, crime, homelessness, prejudice, discrimination, racial tension) [15], and individual level (e.g., illness, behavioural difficulties, drug or alcohol abuse) [16]. Therefore, it is important to create opportunities for children and young people to develop competencies to deal with the social and emotional difficulties they may face during traumatic events. A recent study indicated that youth participation in sports programs is an important resource associated with higher levels of resilience [10]. In this respect, maintaining a regular PA regimen is an important internal resource that can enhance resilience and improve physical and mental health during crises such as the COVID-19 pandemic [10,17].

### Online Mentoring Interventions and Youth Health

According to a review article by Crisp and Cruz (2009) [18], the three main traits of effective mentoring include: (1) a mentoring relationship that emphasizes the individual’s accomplishments and growth; (2) mentoring that offers various forms of support including psychological support and role modelling; and (3) mentoring relationships that are both reciprocal and personal [18].

Youth mentoring is a popular intervention that pairs caring adults with vulnerable young people to promote positive outcomes [19,20]. Effective guidance by mentoring is accomplished through demonstration, instruction, challenge and encouragement on a regular basis over an extended period of time [21]. During this process, the mentor and mentee develop a special bond of mutual commitment. Research indicates that a youth’s sense of connectedness to a caring adult act as a protective factor against a range of risky behaviours. A high-quality youth–mentor relationship is significantly associated with positive social, academic, and health-related behaviours [21,22]. Although formal mentoring has been thought of as a one-to-one, face-to-face relationship, a growing number of programs have begun to experiment with online mentoring relationships [22,23,24]. However, little is known about its relative advantages and disadvantages, or the nature of the relationships formed through this medium. The values of the online mentoring program developers, as well as the objectives of the program, must also be considered [25].

The aim of the present study was to examine online mentoring health intervention (OMHI) among at-risk youth and its association with resilience levels, perceived social support, and reduction in psychological distress and crisis concerns.

## 2. Marerials and Methods

### 2.1. Research Design

This was a non-randomized repeated measures-controlled trial which served as a pilot study. OMHI was created as a collaborative effort by several organizations (five educational institutions and two governmental offices). The program, run by the Israel Ministry of Welfare and Social Affairs and the Department of Labor, involved students from five vocational secondary schools in central Israel. These schools represent a small percentage (about 3%, *n* = 11,600 students) of all secondary school students in Israel. Many of their students have dropped out of mainstream school systems and come from lower socio-economic backgrounds. Their current educational system provides integration for the students on a dual model basis that combines vocational schooling and professional employment [26]. The research staff contacted the principals of the five vocational skills, explained the course and aims of the research study, and requested student participants. Inclusion criteria were: A student enrolled at the vocational school who agreed to participate in the study, whose parents, after receiving an explanation of the study intervention and purpose, signed a form consenting to allow their child’s participation, and who produced a medical certificate allowing them to partake in physical activity. Exclusion criteria were: A student whose parents refused to sign a participation consent form, and/or a student who did not produce a medical certificate that permits sports activities. After the study group of 27 participants was selected, the researchers asked the school principals to select additional youths from the same classes and age- and gender-matched to the study group, to act as the control group.

All 56 students agreed to participate in the study. The students were assured that they had the right to withdraw from the research at any time, that their answers would be kept confidential, and that the questionnaires would be analyzed anonymously.

### 2.2. Study Procedure

All 56 participants completed the study questionnaire twice. Pre-intervention questionnaires were administered before the intervention began in the first week of April 2020, and post-intervention questionnaires were administered following the intervention in the last week of June 2020. The intervention program was delivered with the help of tutors, physical education students from college universities and those with a formal certificate in sports coaching or with a background in sports. It took place online (via zoom or video chat) once per week and included two parts: (1) online physical activity and (2) a ‘heart to heart’ conversation between tutor and student. Each of the two sessions lasted about 30 min, with the exercise tailored to the student’s abilities and desires. The program was accompanied by a multi-professional team (psychologist, professional coordinators, dietitian, health promoters and educational consultants) who provided monthly online and personal guidance to students on how to conduct meaningful conversations with other students, and discussions on issues of risk behaviors, health promotion (importance of exercise, healthy eating, avoidance of risk behaviors and an active and healthy lifestyle), importance of school and learning, goals for the future and dealing with challenges and difficulties. The control group did not participate in the intervention program.

### 2.3. Data Collection and Survey Instrument

The same questionnaire was administered online to the intervention and control groups pre- and post-intervention. The control group did not receive any intervention. The link to the online questionnaire was delivered to the students’ mobile phones. Prompts were given from mentors and the school’s program promoters until all the questionnaires were completed. Each questionnaire took about 20–30 min to complete. The questionnaire was composed of 24 questions and included the following variables: resilience, perceived social support, psychological distress, and crisis concerns.

Ethics approval was received from the Ethics Committee of Ariel University (ref AU-HEA-RT-20210610) before the study commenced.

### 2.4. Independent Variables

#### 2.4.1. Resilience

Resilience was measured by the brief resilience scale (BRS), six statements with which individuals agreed or disagreed, rated on a 5-point scale, ranging from 1 = do not agree at all to 5 = highly agree. When completed, a resilience score between 6 and 30 is generated. Low resilience ranged from a score of 1.00 to 2.99, normal resilience from 3.00 to 4.30, and high resilience from 4.31 to 5.00 [27,28,29]. This score reflects individual feelings of ability and power in the face of difficulties, distress, or trauma. The reliability of this scale in previous research has been shown to be high (α = 0.92) [13].

#### 2.4.2. Perceived Social Support

Perceived social support was evaluated by four items based on the questionnaire operationalized by Zimet et al. [30]. Higher scores reflect perceptions of greater available social support. This inventory was scored on a Likert scale, ranging from 1 = do not agree at all to 5 = highly agree. This tool has demonstrated strong convergent validity and high internal consistency: Cronbach’s alpha ranged from 0.85 to 0.91, and our version ranged from 0.84 to 0.91 [31].

#### 2.4.3. Psychological Distress

Psychological distress level was determined by nine items from the Brief Symptom Inventory (BSI) [32], which concerns anxiety and depression. This inventory was scored by a Likert scale, ranging from 1 = not suffering at all to 5 = suffering very much. Translation and adjustments for Israeli youth demonstrated medium to strong validity and high internal consistency (Cronbach’s alpha ranged from 0.62 to 0.81) [13,33].

#### 2.4.4. Crisis Concerns

Crisis concerns was evaluated by four items that were modified for the COVID-19 crisis, to demonstrate a current and relevant crisis. The questionnaire was based on the political life events scale, including exposure and concerns due to political violence [34]. This inventory was scored on a Likert scale, which ranged from 1 = not worrying at all to 5 = worrying very much. This tool has demonstrated strong convergent validity and high internal consistency: Cronbach’s alpha ranged from 0.80 to 0.88, and our version ranged from 0.75 to 0.88 [33].

### 2.5. Data Analysis

Descriptive statistics (means, SDs, and percentages) of demographic characteristics were calculated using IBM SPSS^®^ Statistics (version 17). The assumptions of normality were also tested. Homogeneity of variance was examined using Levene’s test, with a non-significant test denoting meeting the assumption of equality of variances. The assumption of normality was examined using Shapiro–Wilk’s W test [35], with non-significant results denoting meeting the assumption of normality. As all variables met the assumptions, parametric statistics were used. For the dependent variables, correlations between pre-test and change scores (post-test minus pre-test) were evaluated for each study group separately.

Intra-group changes in the dependent variables from pre- to post-test were examined via paired t-tests and Cohen’s d effect size (mean ∆/SD average from two means). A correction for the dependence among means was conducted using the correlations between the two means following Morris and DeShon’s [36] equation. Generally, values < 0.20 were considered as trivial effect sizes, between 0.20 and 0.50 as small effect sizes, between 0.51 and 0.80 as moderate effect sizes, and >0.80 as large effect sizes. Inter-group differences at both pre- and post- test was examined using independent *t*-tests. Inter-group differences at both pre and post-tests were also evaluated using independent *t*-tests. 

Resilience scores can also be used categorically (low, normal, high resilience [27,28]. Resilience scores’ distribution at pre- and post-test were also presented using box plots. Chi-squared analysis was conducted separately for each study group to examine differences between pre- to post-test in prevalence in each resilience category.

Finally, four separate forward multiple regression analyses (enter method) for factors predicting change from pre- to post-tests in the dependent variables were conducted. Only variables with significant correlations with change scores were included, as well as the study group (intervention vs. control group). In models in which more than one variable was included in the analysis, variables were entered in the order of the correlation’s strength. All four dependent variables (resilience, perceived social support, psychological distress, and crisis concerns) were checked for multicollinearity using variance of inflation factor > 10 [37]. The criterion for inclusion in the model was an alpha level of 0.05, and the exclusion criterion was an alpha level of 0.10.

Only the intervention group and variables that had significant correlations with the dependent variable at pre-test were included in the analyses. Based on these inclusion criteria, in the regression analyses, only a maximum of three predictors were entered. Post-hoc power analysis using the study’s average effect sizes (intervention and control group average Cohen’s d effect size = 0.65) for multiple regression showed that for the study’s primary outcome measures, the power achieved was 0.88 with the three predictors. Power analysis was conducted using G*Power 3.0.10. In all statistical analyses, *p*-values of <0.05 were considered as statistically significant [38].

#### Ethical Considerations

Ethics approval was received from the Ethics Committee of Ariel University (ref AU-HEA-RT-20210610) before the study commenced. The head of therapeutic services at the Israel Ministry of Welfare and Social Affairs and the Department of Labor approved the study. A preliminary letter regarding the survey was sent to the parents of the students. They were asked to confirm their children’s participation. On the day of the survey, it was made clear to the students that the questionnaire was anonymous and their names should not be written.

## 3. Results

### 3.1. Descriptive Results

The 56 participants had a mean age of 16.18 + 0.83 and 16.62 + 0.82 years in the intervention and control groups, respectively (independent test statistic *t* = 1.97, *p* = 0.06). In both groups, most participants were males (intervention group: *n* = 23 males, 85.2% of the sample; control group: *n* = 28 males, 96.6% of the sample).

### 3.2. Main Analyses

#### 3.2.1. Correlations between Dependent Variables

Several statistically significant (0.0002–0.049) correlations were observed between the dependent variables (pre-test with change scores), in both study groups. More specifically, pre-test resilience scores significantly correlated with resilience change scores (intervention group: −0.675; control group: −0.437; *p* < 0.001) and psychological distress (intervention group: 0.400; control group: 0.389; *p* < 0.001). Pre-test crisis scores significantly correlated only with psychological distress (intervention group: −0.354; control group: −0.365; *p* < 0.001). Pre-test psychological distress significantly correlated with several change scores, namely, resilience (intervention group: −0.355; control group: −0.432; *p* < 0.001) and psychological distress (intervention group: 0.593; control group: 0.399; *p* < 0.001). Finally, pre-test social support did not significantly correlate with any of the change scores (Table 1).

There were statistically significant (*p* < 0.001–0.03) differences between the groups at pre-test in two of the four dependent variables. Specifically, compared to the control group, the intervention group presented higher psychological distress and crisis levels (*p* < 0.001 and 0.03, respectively) (Table 1). At post-test, significant inter-group differences were observed only in one variable: crisis concerns (intervention group: 13.51 (SD 3.82), control group: 8.88 (SD 3.87); *p* < 0.001 (Table 2).

From pre- to post-test, the intervention group experienced statistically significant changes in all study variables (*p* < 0.001–0.02) (Table 2). Three of the observed changes were positive and suggested an improvement in the participants’ state (i.e., increased resilience, increased social support, and decreased psychological distress). In terms of effect size, the aforementioned positive observed changes were all substantial (Cohen’s d > 0.80) (Table 3). The negative change was in crisis concerns. This change effect size was moderate (Cohen’s d = 0.64; Table 3). The intervention group presented one positive change in nature: an increase in perceived social support (Table 2), though the effect size suggests a small difference (Cohen’s d = 0.40) (Table 3).

The vertical lines extend from the minimum to the maximum value, excluding outside and far out values, which are displayed as separate points. An outside value is defined as a value that is smaller than the lower quartile minus 1.5 times the interquartile range, or larger than the upper quartile plus 1.5 times the interquartile range (inner fences). A far-out value is defined as a value that is smaller than the lower quartile minus 3 times the interquartile range, or larger than the upper quartile plus 3 times the interquartile range (outer fences).

Figure 1 shows the distribution of resilience scores, namely, the number and percentage of participants with low, normal, and high resilience. The control group did not present any significant changes from pre- to post-tests in the number and percentage of participants in each resilience category. The intervention group presented a significant decrease in the number of participants with low resilience (pre-test, 66.7% of the sample; post-test, 11.1% of the sample; *p* < 0.0001) and a significant increase in the percentage of participants with normal resilience (pre-test, 33.3% of the sample; post-test, 88.9% of the sample; *p* < 0.0001).

#### 3.2.2. Prediction of Change Scores (Pre-Test to Post-Test)

Table 4 presents the results of the regression analysis for the prediction of change scores from pre- to post-test in the four dependent variables. Overall, the models explained 1.4% (change in social support) to 49% (change in crisis) of the variability of change scores. Greater resilience at pre-test significantly predicted a greater resilience change score (unstandardized B coefficient = 0.65). Greater psychological distress at pre-test significantly predicted greater change in psychological distress (unstandardized B coefficient = 0.45). Greater crisis at pre-test significantly predicted a greater crisis change score (unstandardized B coefficient = 0.58). Social support at pre-test did not predict any of the change scores. The intervention group was a significant predictor of the change scores of resilience and crisis (unstandardized B coefficient = 0.54 and 3.333, respectively), with belonging to the intervention group predicting greater changes than belonging to the control group. 

## 4. Discussion

Few studies have been conducted on the impact of online mentoring intervention programs that aim to encourage interpersonal bonding and physical and mental health in times of crisis and distress among at-risk youth [24,25].

The results of the current study showed that the OMHI would lead to an increased resilience level among at-risk youth. The level of resilience in those participating in the intervention program was lower than in the control group at pre-test but similar at post-test. In terms of effect size, the degree of change from pre- to post-test in resilience was greater in the intervention group vs. the control group. The research results are consistent with findings of other studies and support the need to promote PA in combination with online mentoring for at-risk youth to help them cope with crises and states of distress and trauma [9,10]. Resilience can be developed through building internal strength and external resources [12,13]. For example, one study that investigated youth participation in online sports during the COVID-19 pandemic showed elevated resilience: those who participated in structured online PA programs during lockdown periods were significantly more resilient and physically active, had higher self-related health, were more satisfied with life, and were better able to cope during the pandemic compared to non-participants [9].

The results of our study also showed that participation in OMHI led to increased levels of perceived social support. In both groups, there was a significant difference between the reported level of perceived social support measured pre- and post-program. However, the extent of change (effect size) in the intervention group was greater than in the control group. The increase in the level of perceived social support is of special importance as previous research has indicated that a young person’s sense of connection to a caring adult acts as a protective factor against a range of risky behaviours. Moreover, high-quality youth–mentor relationships have been significantly associated with positive social- and health-related behaviours [22,23,24].

Research findings support that participation in OMHI would reduce the intensity of psychological distress. In the intervention group, there was a significant difference between the reported level of psychological distress measured pre- and post-program. No significant changes were observed in the control group. These results are in line with studies showing both direct and indirect relationship between PA and mental health [17,18].

Finally, research findings could not verify that participation in OMHI would reduce the sense of crisis concerns, distress, and tension. In the intervention group, despite the intervention, a significant increase in crisis-concern level was observed from pre- to post-test. The control group did not show any difference in the reported level of crisis concerns before and after the intervention; however, their initial crisis levels at pre-test were significantly lower than those of the intervention group. Recent studies have shown that youth have experienced high rates of anxiety and depression symptoms during the COVID-19 pandemic [39,40]. The resulting social isolation and economic uncertainty have led to a significant increase in mental health concerns [41,42]. Recent studies have also revealed that young people’s greatest worry during lockdown was being socially disconnected, a state that is associated with higher levels of anxiety and depressive symptoms, and lower levels of life satisfaction [43,44]. Despite PA and mentoring support, OMHI did not improve such concerns.

This study had both strengths and limitations. One main strength was that this study is among the first in Israel to investigate the connection between PA, resilience, perceived social support, psychological distress, and concerns among at-risk youth. Additionally, as the activities were completed in couples (not individually), the intervention program had a high chance of being successful A final factor that could have contributed to the program’s success is the fact that activities were conducted online and not in a traditional classroom setting.

There were also several limitations to this study. Our findings were based on a self-reporting questionnaire, which could cause bias. Additionally, the cross-section study design with convenience sampling approach limits the generalizability of the results. Future studies should investigate online learning in large sample sizes and in different countries. A final limitation includes the use of just one tool (the questionnaire). Future studies should focus utilize various tools (e.g., observations or semi-structured interviews), which could be useful for detecting more complex and deeper insights.

## 5. Conclusions

The results of this study indicated that at-risk youth participation in an online PA and “heart-to-heart” mentoring intervention program was an important and effective resource for increasing resilience, elevating social support, and diminishing psychological distress and crisis concerns. This type of program can be an effective strategy in health promotion through multi-faceted interventions for at-risk youth during times that are challenging either on an individual basis or on a wider scope. Research study on youth’s use of online mentoring in physical and emotional health promotion programs is lacking, specifically those that focus on direct observation and mixed measurement methods that might enable a wider understanding of the benefits, disadvantages, and feasibility of such interventions during regular, routine and non-crisis periods.

## Figures and Tables

**Figure 1 children-09-01704-f001:**
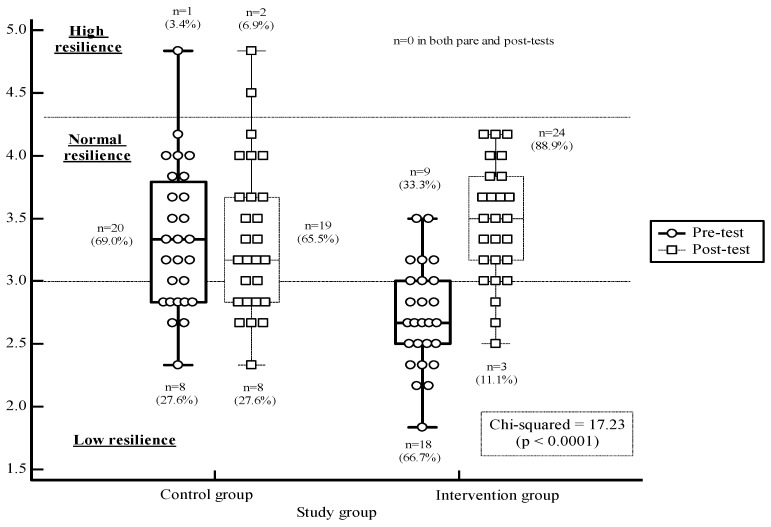
Resilience at pre- and post-tests: prevalence of low, average, and high resilience.

**Table 1 children-09-01704-t001:** Correlations between pre-test and change scores (Pearson correlations).

Variables	Resilience—Change Score		Social Support—Change Score		Crisis—Change Score		Psychological Distress—Change Score	
	Intervention group	Control group	Intervention group	Control group	Intervention group	Control group	Intervention group	Control group
	(n = 27)	(n = 29)	(n = 27)	(n = 29)	(n = 27)	(n = 29)	(n = 27)	(n = 29)
	r	r	r	r	r	r	r	r
	(*p* value)	(*p* value)	(*p* value)	(*p* value)	(*p* value)	(*p* value)	(*p* value)	(*p* value)
Resilience—Pre−test								
	−0.675	−0.437	−0.152	0.196	0.180	0.123	0.400	0.389
	(0.0002)	(0.049)	(0.457)	(0.346)	(0.993)	(1.000)	(0.037)	(0.036)
Social support—Pre−test								
	−0.021	−0.171	0.177	−0.312	0.001	−0.249	−0.162	−0.22
	(0.919)	(0.413	(0.121)	(0.378)	(0.220)	(0.120)	(0.279)	(0.125)
Crisis—Pre−test								
	−0.034	−0.045	−0.128	−0.115	−0.354	−0.365	0.156	0.230
	(0.916)	(0.9)	(0.118)	(0.119)	(0.009)	(0.008)	(0.111)	(0.446)
Psychological distress—Pre−test								
	−0.355	−0.432	−0.200	0.079	0.001	−0.090	0.593	0.399
	**(0.009)**	**(0.027)**	(0.700)	(0.317)	(0.661)	(1.000)	**(0.001)**	**(0.04)**

Notes: Change score: post-test—pre-test; Significant at the *p* < 0.001 level. Inter-Group Analysis: Differences Between Intervention and Control Groups at Pre- and Post-Test.

**Table 2 children-09-01704-t002:** Inter- and intra-group differences pre- and post-test.

	Intervention Group (*n* = 27)			Control Group (*n* = 29)			Inter-Group Analysis	
Variables	Pre-test	Post-test	Within group	Pre-test	Post-test	Within group	Pre-test	Post-test
	mean	mean	statistic t	mean	mean	statistic t	statistic t	statistic t
	(SD)	(SD)	(*p* value)	(SD)	(SD)	(*p* value)	(*p* value)	(*p* value)
Resilience								
	2.71	3.45	5.80	3.34	3.35	0.32	4.63	−0.69
	(0.4)	(0.46)	(>0.001)	(0.57)	(0.61)	(0.74)	(<0.001)	(0.49)
Perceived social support								
	17.34	19.74	3.47	18.14	19.32	5.11	1.17	−0.45
	(2.07)	(3.7)	(0.001)	(2.81)	(3.05)	(<0.001)	(0.24)	(0.64)
Crisis concerns								
	11.19	13.51	2.40	8.88	8.89	0.02	−1.19	−4.41
	(3.68)	(3.82)	(0.02)	(3.95)	(3.87)	(0.69)	(0.03)	(0.001)
Psychological distress								
	2.55	1.85	−3.67	1.59	1.60	1.54	−5.59	−1.45
	(0.66)	(0.67)	(0.001)	(0.60)	(0.61)	(0.13)	(<0.001)	(0.15)

Note: SD = standard deviation. Inter-Group Analysis: Changes from Pre- to Post-Test in Intervention and Control Groups.

**Table 3 children-09-01704-t003:** Cohen’s d for the mean difference (repeated measures).

	Intervention Group (*n* = 27)		Control Group (*n* = 29)	
Variables	Cohen’s d	95% CI	Cohen’s d	95% CI
Resilience				
	1.88	1.07 to 2.81	0.01	−0.05 to 0.08
Perceived social support				
	1.22	0.52 to 1.97	0.40	0.23 to 0.57
Crisis concerns				
	0.64	0.08 to 1.43	0.00	0.00 to 0.00
Psychological distress				
	−1.03	−1.67 to −0.42	0.03	0.00 to 0.08

Notes: CI = confidence interval. Cohen’s d is based on a single pooled standard deviation and was corrected for dependence between means using Morris and DeShon’s equation; substantial differences (>0.80) are denoted in dark gray cells, moderate differences (0.51–0.80) are denoted in light gray cells, and trivial (<0.20) and small differences (0.21–0.50) are denoted in white cells.

**Table 4 children-09-01704-t004:** Variables predicting change in dependent variables.

Dependent Variables	Independent Variables	Unstandardized B Coefficient	Unstandardized Standard Error	*t*	*p*
Resilience–change score					
	Constant	1.17			
	Resilience—pre-test	0.65	0.12	5.07	<0.001
	Psychological distress—pre-test	−0.00	0.10	−0.07	0.940
	Intervention group (In comparison to control group	0.54	0.16	3.29	0.001
R^2^ = 0.373; Adjusted R^2^ = 0.333; F ratio = 9.335; *p* = 0.001					
Social support–change score					
	Constant	19.32			
	Intervention group (in comparison to control group)	0.41	0.91	0.45	0.640
R^2^ = 0.0003; Adjusted R^2^ = 0.014; F ratio = 0.21; *p* = 0.64					
Crisis– Change score					
	Constant	3.65			
	Crisis—pre-test	0.58	0.12	4.89	<0.001
	Intervention group (In comparison to control group)	3.33	0.94	3.54	0.009
R^2^ = 0.51; Adjusted R^2^ = 0.49; F ratio = 25.58; *p* < 0.0001					
Psychological distress– Change score					
	Constant	0.99			
	Resilience—pre-test	−0.02	0.16	−0.14	0.88
	Psychological distress—pre-test	0.45	0.13	3.37	0.001
	Intervention group (In comparison to control group)	−0.18	0.21	−0.83	0.40
R^2^ = 0.22; Adjusted R^2^ = 0.17; F ratio = 4.68; *p* = 0.006					

Note: Only intervention group and variables that had significant correlations with the dependent variable at pre-test were included; in models in which more than one variable was included in the analysis, variables were entered in order or by correlation strength.

## Data Availability

The author is the sole person who conceived, did the research and wrote the article.

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
