# Peer review of "The Effects of a Physical Activity Online Intervention Program on Resilience, Perceived Social Support, Psychological Distress and Concerns among At-Risk Youth during the COVID-19 Pandemic"

_children, 2022, doi:10.3390/children9111704_

Round 1
Reviewer 1 Report
I have read this paper with interest - it is so important to support young students during difficult COVID-times - I therefore welcome the subject of the paper. I like the idea of combining physical activities with (verbal) support offered by a mentor. My suggestions/questions are the following:
With regard to methods:
- it is unclear whether the control group did receive any intervention, or conversations at all, or nothing, and they participated in a pre and post test?;
- it still is unclear how exactly the students that participated in the intervention were recruited - which factors influenced the choice to refer them?;
- 30 minutes that is very brief - how much time exactly was spent on physical activities, and how much time on talking? What kind of physical activities were organized? Has it been the same offer to all students, weekly?
- Related - was treatment fidelity checked? in other words - have sessions been audio- or videotaped and observed by researchers?; Was the intervention offered as expected?
- with regard to title and introduction - the introduction starts with underlining the importance of physical activity - yet no reference to it has been made in the title - maybe adjust the title to make it more congruent with the content of the paper?
- Repetition: Sentences part of 2.4 has been repeated in the text.
- Lay-out: Tables should be formatted according to APA?, or is this the lay-out by Children?
I hope these recommendations will be feasible; the paper of supporting students during COVID by offering an easily accessibel intervention may be very informative.
Author Response
Comments and suggestions for reviewer number 1:
I have read this paper with interest - it is so important to support young students during difficult COVID-times - I therefore welcome the subject of the paper. I like the idea of combining physical activities with (verbal) support offered by a mentor.
With regard to methods:
- it is unclear whether the control group did receive any intervention, or conversations at all, or nothing, and they participated in a pre and post test?
Section 2.4 of the methods states: “The same questionnaire was administered online to the intervention and control groups pre- and post-intervention.” We added the following sentence: “The control group did not receive any intervention.”
it still is unclear how exactly the students that participated in the intervention were recruited - which factors influenced the choice to refer them?
Section 2.1 in the methods includes the inclusion criteria, which states: “A student enrolled at the vocational school who agreed to participate in the study, whose parents, after receiving an explanation of the study intervention and purpose, signed a form consenting to allow their child’s participation, and who produced a medical certificate allowing them to partake in physical activity.”
There were 27 students in the intervention group and 29 students in to the control group.
30 minutes that is very brief - how much time exactly was spent on physical activities, and how much time on talking? What kind of physical activities were organized? Has it been the same offer to all students, weekly?
The program includes 2 parts. In the first part, physical activity (e.g., walking, exercise, running) was conducted. In the second part, the student and mentor had a conversation about healthy lifestyle, aims for the following year, etc.; each one of these parts was 30 minutes.
In section 2.3 of the methods, we have edited the text to now say: “It took place online (via zoom or video chat) once per week and included two parts: 1) online physical activity and 2) a 'heart to heart' conversation between tutor and student. Each of the two sessions lasted about 30 minutes,”
- Related - was treatment fidelity checked? in other words - have sessions been audio- or videotaped and observed by researchers? Was the intervention offered as expected?
The research tools included only survey; sessions were not recorded/observed by the researchers.
- with regard to title and introduction - the introduction starts with underlining the importance of physical activity - yet no reference to it has been made in the title - maybe adjust the title to make it more congruent with the content of the paper?
We have changed the title to:
“The Effects of a Physical Activity Mentoring Online Intervention Program on Resilience, Perceived Social Support, Psychological Distress and Concerns Among At-Risk Youth During the COVID-19 Pandemic”
- Repetition: Sentences part of 2.4 has been repeated in the text.
Thank you for pointing this out; we have combined the section into one.
- Lay-out: Tables should be formatted according to APA?, or is this the lay-out by Children?
The tables have been formatted according to Children.
I hope these recommendations will be feasible; the paper of supporting students during COVID by offering an easily accessibel intervention may be very informative.
Thank you very much for your review; this program continues to be a major component of the students at the participating schools and universities.

Reviewer 2 Report
Dear Authors!
Thank you so much for the opportunity to review your manuscript. It is interesting from the practical point of view.
My suggestions: 1) Introduction should be more concise, and focused on the manuscript's aim. 30 references for introduction is too much. I think authors can transfer some statements and references from introduction to discusiion. Now discussion is short and might be more
Author Response
Comments and Suggestions for Author number 2:
Thank you so much for the opportunity to review your manuscript. It is interesting from the practical point of view.
Thank you very much for your review; this program continues to be a major component of the students at the participating schools and universities.
My suggestions: 1) Introduction should be more concise and focused on the manuscript's aim.
We have shortened the introduction so that now it should be more focused on the aim.
30 references for introduction is too much. I think authors can transfer some statements and references from introduction to discussion.
We agree that the number of references included in the introduction is indeed a bit high – we have taken some out.
Now discussion is short and might be more
We added more information to the discussion.
